# Exploring Intention toward Using an Electric Scooter: Integrating the Technology Readiness and Acceptance into Norm Activation Model (TRA-NAM)

**Chien-Wei Ho [1],\* and Chi-Chuan Wu [2]**

1   School of Management, National Taiwan University of Science and Technology, Taipei 106335, Taiwan
2   Department of Business Management, Tatung University, Taipei 104, Taiwan; ccwu@ttu.edu.tw
\*   Correspondence: chienweiwhite@gmail.com

**Abstract:** The issues of renewable energy, energy crisis, and carbon reduction have caught people's attention all over the world, and governments have put forth greater effort to proactively solve these problems. Electric transportation not only benefits the environment, but can also utilize renewable energy to prevent an energy crisis. Based on previous theoretical strands of the literature, this research integrates the technology readiness and acceptance model (TRAM) into the norm activation model (NAM) and proposes an integrated model denoted as TRA-NAM. It takes TRA-NAM as our theoretical foundation and aims to explore the effect of technology readiness and awareness of consequence on the intention toward using an electric scooter (ES). The results display that technology readiness positively influences perceived usefulness and perceived ease of use and further improves consumers' intention toward adopting ES. In addition, personal norm mediates the relationship between awareness of consequence and intention to adopt ES. This study offers the integrated TRAM-NAM model in order to understand the crucial factors affecting consumers' intention to adopt electric vehicles (EVs). Overall, this research fills the gap in the field of government policies and transportation and proposes ponderable suggestions, in particular that if they want to encourage or attract consumers to drive an ES, they should not overlook the effect of technology readiness and awareness of consequence.

**Keywords:** technology readiness; awareness of consequence; personal norm; electric scooter



## 1. Introduction

The heavy use of non-renewable resources such as fossil fuels is wreaking havoc on nature and leading to climate change [1,2]. IEA [3] reported that over 49.9% of transportation are powered by fossil fuels, which are responsible for 8.4% of the total $CO_2$ emission all over the world. Due to the growing issues of extreme climate change, global warming, $CO_2$ emissions, and energy crises around the world, governments and transportation industries have put much more effort into developing various forms of electric transportation. Electric transportation not only offers opportunities for introducing renewable energy (such as solar power and wind power), but also helps diminish $CO_2$ emissions and air pollution.

There are some studies discussing the issues of electric transportation. For example, based on environmental concerns, Joanes [4] found that problem awareness and self-efficacy affect personal norm, and that via justice and effectiveness, personal norm stimulates the purchase and use of EVs. He and Zhan [5] regarded that personal norms improve people's intention to adopt EV. Nevertheless, there is still a gap concerning a comprehensive research framework to examine the effects of crucial antecedents on using electric transportation, such as consumers' propensity to do so.

Moreover, IRENA [6] reports on the growth and popularity of two-wheel vehicles in Africa and Asia, with more than 80% of households surveyed in Indonesia, Malaysia, Thailand, and Vietnam owning a gas-powered motorcycle [7]. Therefore, it is very crucial

for governments and transportation industries to encourage people to use electric two-wheel motored vehicles to deal with energy crises and pollution. However, most studies and reports focus on electric cars with a shortage of research on electric two-wheeled transportation.

This research therefore sheds light on the required theories to understand the effects of the psychological process on intention to adopt ES. More specifically, this research integrates the technology readiness and acceptance model (TRAM) into the norm activation model (NAM) and proposes a combined model called TRA-NAM. According to TRA-NAM, we aim to present how technology readiness can, through perceived usefulness and ease of use, affect one's intention to adopt ES. Furthermore, this research examines the relationships among awareness of consequences, personal norm, and intention to adopt ES simultaneously. Based on the research framework in Figure 1, this study contributes to academia by proposing an integrated model to explain consumer behavior and intention towards green vehicles and also provides suggestions for governments that want to develop and enhance the ES usage rate in the future.

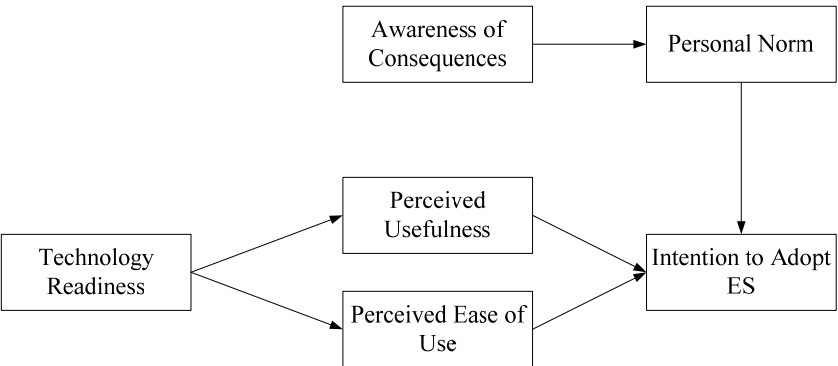

**Figure 1.** Research framework.

The rest of this study is organized as follows. First, this study briefly explains the research background and motivation. Second, we introduce TRAM and NAM and then propose the hypotheses among technology readiness, perceived usefulness, perceived ease of use, awareness of consequences, personal norm, and intention to adopt ES based on previous studies. Third, this study explains the procedure of data collection as well as the measurements for each variable. Fourth, we conduct an analysis on reliability and validity, goodness-of-fit, and hypotheses' tests. Finally, based on the results, we offer a conclusion, propose contributions for academia and the real world, and provide limitations and future directions for this research topic.

## 2. Materials and Methods

### 2.1. Technology Readiness and Acceptance Model (TRAM)

Based on the theory of reasoned action (TRA) developed by Ajzen and Fishbein [8], Davis [9] proposed the technology acceptance model (TAM) to predicting consumer behavior when adopting technology and included two key beliefs that determine attitude, perceived usefulness, and perceived ease of use. Perceived usefulness is defined as the degree to which users who adopt a specific technology can improve their performance, whereas perceived ease of use refers to the degree to which users regard that adopting a specific technology can be easy [10]. TAM is regarded as one of the most widely used models [11], and Kim et al. [12] and Lam et al. [13] apply it to explore the opinions of hotel industry employees on their intention of using technology. In addition, TAM has been employed to examine the intention of using Advanced Driver Assistance Systems [14] as well as drivers' intentions to adopt EVs [15].

Parasuraman [16] proposed a construct of technology readiness and defined it as people's propensity to embrace and adopt new technologies to accomplish goals in their

home life and at work [16] (p. 308). Technology readiness is measured by TRI and contains four factors: optimism, innovativeness, discomfort, and insecurity. Optimism means a positive view of technology; innovativeness refers to a tendency to be the first to use a new technology; discomfort is regarded as having a need for control and a sense of being overwhelmed; insecurity means distrusting a technology for security and privacy reasons [17] (p. 207). Technology readiness is widely used to understand the adoption of technologies among users [18].

The impact of technology readiness on TAM has also been recently considered. Lin et al. [19] combined technology readiness and TAM to develop an integrated TRAM (technology readiness and acceptance model). TRAM allows us to understand the personal traits of a consumer and how to, via used experience, influence the adoption intention or behavior of a technological product. For example, Chen et al. [20] incorporated an individual's health consciousness into TRAM and tried to predict their attitude and intention toward downloading and utilizing a fitness app.

People with high technology readiness tend to try any new technology and have a belief that technology provides them with increased control, flexibility, and efficiency in their lives [21] (p. 34). Those scoring high in technology readiness are likely to accept technology more freely, because they will not focus on the drawbacks. Moreover, high technology readiness people have a positive impression of its usefulness in general, despite its potential value being uncertain and not obvious [17]. Sun et al. [11] regarded that technology readiness positively affects perceived ease of use and usefulness. Therefore, we expect that technology readiness has a positive influence on perceived ease of use as well as usefulness of ES, and thus propose the following hypothesis:

**Hypothesis 1a.** *High technology readiness leads to higher perceived ease of use of ES.*

**Hypothesis 1b.** *High technology readiness leads to higher perceived usefulness of ES.*

Based on the TAM model, one study reported that perceived ease of use and perceived usefulness positively affect use intention [22]. When a technology is regarded as easy to use and offers usefulness, then people will more likely exhibit a behavior intention on a specific technology. If people regard that ES is easy to control and actually offers benefits for their lives, then this will build a good impression toward ES technology and further influence their intention to adopt it. Lin et al. [23] proposed the TRAM model and provided support that perceived ease of use and perceived usefulness significantly influence use intention. Hence, this study develops the following hypotheses that perceived ease of use has a positive relationship with intention for adopting ES and perceived usefulness also has a positive relationship with the same intention:

**Hypothesis 2.** *Perceived ease of use positively influences the intention for adopting ES.*

**Hypothesis 3.** *Perceived usefulness positively influences the intention for adopting.*

### 2.2. Norm Activation Model (NAM)

Schwartz [24] proposed the concept of NAM in the context of altruistic behavior and provided support that such behavior is affected by personal norm. NAM includes three major variables to predict pro-social behavior [25]. The first, awareness of consequences (AC), refers to whether someone is aware of the negative consequences for others or for other things one values when not acting pro-socially [25] (p. 426). The second is ascription of responsibility (AR), which means whether someone is aware of the negative outcomes for others' values when not acting pro-socially [25] (p. 426). The third is personal norm (PN) and is described "as feelings of moral obligation not as intention" [24] (p. 277). De Groot and Steg [25] found that personal norm plays a crucial role in mediating the relationship between ascription of responsibility and pro-social intention or behavior, and ascription

of responsibility is predicted by awareness of consequences. NAM has recently been applied in predicting environmentally friendly behavior, such as attending an environmentally responsible convention [26], energy conservation behavior [27], waste separation intention [28], and intention to visit restaurants featuring organic menu items [29].

Using a conventional gas scooter might lead to negative consequences, e.g., air pollution, $CO_2$ emissions, and consumption of oil resources. If citizens are aware that using conventional scooters causes an exhaustion of energy and has a serious negative effect on the environment, then this is likely to elicit their moral obligation to avoid negative consequences. On the contrary, if citizens are not aware of these negative consequences, then they are not likely to develop personal norm of use ES. Prior literature also provided support that an awareness of consequences has a significant effect on personal norm. For instance, Zhang et al. [30] proposed that there is a positive relationship between awareness of consequences and personal norm. Our study expects that awareness of consequences has a positive influence on personal norm and hence proposes the following hypothesis:

**Hypothesis 4.** *Awareness of consequences positively affects personal norm.*

When people feel a moral obligation to conduct pro-social acts, they will be motivated to exhibit pro-social behaviors that align with certain value systems. Therefore, people with personal norm resulting from the negative outcomes of conventional scooters will feel an obligation to do something to abate those negative consequences. They will then have a greater motivation to adopt pro-environmental products. In other words, if people have high personal norm, then instead of using conventional scooters, they will have more motivation to engage in using an ES. A few studies have also provided support that personal norm influences environmentally friendly behavior. Wang et al. [28] explored waste separation behavior in China and found that personal norm significantly influences such intention by residents. In addition, Song et al. [31] assert that when people have a personal norm, it will promote consumption behavior toward energy-saving appliances. Taken together, this study expects that personal norm positively influences intention to adopt ES and offers the following hypothesis:

**Hypothesis 5.** *Personal norm positively affects intention toward adopting ES.*

### 3. Method

#### 3.1. Data Collection and Procedure

In response to environment crisis and energy issues due to the major form of transportation in the country, i.e., scooters, the Taiwanese government has put much more effort into supporting the ES industry in recent years. ITRI (Industrial Technology Research Institute, Zhudong Township, Taiwan) has regarded that ES is gradually playing an important role in the nation's scooter industry. In addition, the usage rate (electric scooter number/total scooter number) of ESs in Taiwan increased from 0.55% in 2016 to 2.6% in 2019 in, showing that people have come to gradually accept ESs. As mentioned above, this study views Taiwan as source to collect data, and aims to examine the relationships among technology readiness, perceived usefulness, perceived ease of use, awareness of consequences, personal norm, and intention for adopting ES. Before respondents filled in the questionnaire, we assured them that their information and answers would be collected solely for academic use and would remain confidential. The first section of questionnaire is a statement to introduce the research objective and the background information of ESs. The next section measures the variables in this research. Finally, the last section collects demographic information of participants. In order to avoid the common method bias, this study disarranges the order of variables in the second section. Furthermore, this study excludes those respondents without a scooter license and those who have not ridden a scooter in the past two years in order to make guarantee the quality of the questionnaire.

Prior to the formal data collection process, this study conducted pre-test procedures to assure that our questionnaire is easy to understand and measures the research variables properly. First, we took the suggestion from Brislin [32] to confirm that the meanings of the survey items were similar to the original version. Second, this research invited students, academic, and employees working in the ES industry to fill in the questionnaire. The coefficient alphas of variables (technology readiness, perceived usefulness, perceived ease of use, intention to adopt ES, awareness of consequences, and personal norm) are 0.918, 0.972, 0.940, 0.945, 0.901 and 0.810 respectively, proving that all variables had a great reliability.

Excluding those who had no scooter license, had not ridden a scooter in the past two years, and incomplete answers, we left out 73 out of the 300 questionnaires (effective rate is 75.7%). Among survey respondents, males account for 60.4% and females account for 39.6%; the majority of respondents are between the ages of 18 and 25 (46.7%), followed by the ages between 26 and 35 (17.6%) as well as between 46 and 55 (16.3%). Most of our respondents have a bachelor's degree (48.5%) and the majority of the respondents' income is between NT$20,000 and NT$30,000 (42.7%) per month.

### 3.2. Measurement

This study includes two exogenous variables (technology readiness and awareness of consequences) and four endogenous variables (perceived usefulness, perceived ease of use, personal norm, and intention to adopt ES) and aims to understand how exogenous variables influence endogenous variables. We adopt five-point Likert scales (1 = 'strongly disagree'; 5 = 'strong agree') in our questionnaire.

Technology readiness. This study adopts second-order confirmatory factor analysis (CFA) to measure technology readiness. We view optimism and innovativeness as our first-order constructs to measure technology readiness and adopt five-item scales ("Technology makes me more efficient in my occupation"; "Technology gives me more freedom of mobility"; "Learning about technology can be as rewarding as the technology itself"; "I find new technologies to be mentally stimulating"; and "I prefer to use the most advanced technology available") to measure optimism and five-item scales ("I can figure out new high-tech products and services without any help"; "Others come to me for advice on new technology"; "I am among the first in my circle of friends to acquire new technology"; "I have fewer problems than others in making technology work"; and "I keep up with the latest technological development that I am interested in") to measure innovativeness [16,30,33].

Perceived usefulness. This study uses a three-item scale to measure perceived usefulness [33,34], and the items involve: "Using an electric scooter in my job/life will increase my productivity"; "Using an electric scooter in my job/life will make me more effective"; and "I find an electric scooter to be useful in my job/life".

Perceived ease of use. We use three items developed by Davis [22] to measure perceived ease of use, and they include: "My experience with using an electric scooter is better than what I expected"; "The service level provided by an electric scooter is better than what I expected"; and "Overall, most of my expectations from using an electric scooter are confirmed".

Awareness of consequences. Zhang et al. [35] developed three items to measure awareness of consequences, which we adopted: "Riding a conventional scooter causes exhaustion of energy"; "Riding a conventional scooter contributes to environment damage"; "Riding a conventional scooter has an effect on global warming".

Personal norm. Jansson [36] measured personal norm with a four-item scale, which we adopted: "I feel a moral obligation to conserve fossil fuels and protect the environment no matter what other people do"; "I feel that it is important to travel as little as possible by car using fossil fuels"; "I feel a moral obligation to ride an electric scooter instead of a conventional scooter"; "People like me should do everything they can to decrease the use of fossil fuels such as oil/petrol/diesel".

Intention to adopt ES. This research measures intention to use an ES with three items from Barbarossa et al. [37], and they include: "The next time I buy a scooter, I will consider buying an electric scooter"; "I expect to drive an electric scooter in the near future"; and "I intend to drive an electric scooter in the near future".

## 4. Results

### 4.1. Reliability and Validity

Based on Table 1, we can see the description statistics and the results of the Pearson correlation analysis. Most of our respondents have higher technology readiness. In addition, the values of correlation coefficient represent that all the variables have significant correlations. Before conducting the hypothesis test, this study has to first confirm reliability. Reliability is a crucial issue in questionnaire research and refers to the consistency of a variable. In other words, it shows that the researchers could expect respondents to give the same response to an item on a variable. We use component reliability (CR) to verify reliability, and the results of CR are shown in in Table 2. In Table 2, the CR values of technology readiness, perceived usefulness, perceived ease of use, and intention to adopt ES, awareness of consequences, and personal norm are 0.722, 0.957, 0.937, 0.924, 0.898, and 0.866, respectively. The fact that all are higher than 0.7 indicates that each variable has great reliability [38].

**Table 1.** Description statistics of each variable.

|     | M     | SD    | TR       | PU       | PEU      | INT      | AC       | PN    |
|-----|-------|-------|----------|----------|----------|----------|----------|-------|
| TR  | 4.031 | 0.600 | 0.756    |          |          |          |          |       |
| PU  | 3.328 | 1.095 | 0.339 ** | 0.939    |          |          |          |       |
| PEU | 3.467 | 0.966 | 0.379 ** | 0.729 ** | 0.912    |          |          |       |
| INT | 3.348 | 0.985 | 0.381 ** | 0.733 ** | 0.796 ** | 0.896    |          |       |
| AC  | 4.171 | 0.808 | 0.197 ** | 0.250 ** | 0.317 ** | 0.287 ** | 0.863    |       |
| PN  | 4.090 | 0.710 | 0.363 ** | 0.466 ** | 0.438 ** | 0.479 ** | 0.569 ** | 0.789 |

Notes: TR = Technology Readiness, PU = Perceived Usefulness, PEU = Perceived Ease of Use, INT = Intention to Adopt ES, AC = Awareness of Consequences, PN = Personal Norm, M = Mean, SD = Standard Deviation. The figures in diagonal text represent the root of AVE values for each construct. ** $p < 0.01$.

Moreover, this study has to confirm validity, defined as a concept or construct is accurately measured in a survey study. This research adopts the convergence and discriminant tests to verify validity. For convergent validity, when average variance extracted (AVE) is bigger than 0.5 and CR is bigger than 0.7, this represents great convergent validity [38]. For discriminant validity, Henseler et al. [39] regarded that each variable's squared root of AVE must be higher than the correlation with other variables in the research model. The values in the diagonal text of Table 1 display that all squared root values of AVE are bigger than the correlation with other variables, thus supporting that this study has great discriminant validity.

**Table 2.** Index of reliability and validity.

| Variable | Item | Factor Loading | CR | AVR |
|---|---|---|---|---|
| Technology Readiness | Optimism | 0.613 | 0.722 | 0.572 |
| | Innovativeness | 0.877 | | |
| Perceived Usefulness | PU1 | 0.944 | 0.957 | 0.882 |
| | PU2 | 0.967 | | |
| | PU3 | 0.905 | | |
| Perceived Ease of Use | PEU1 | 0.887 | 0.937 | 0.832 |
| | PEU2 | 0.959 | | |
| | PEU3 | 0.889 | | |
| Intention to Adopt ES | INT1 | 0.902 | 0.924 | 0.803 |
| | INT2 | 0.923 | | |
| | INT3 | 0.862 | | |
| Awareness of Consequences | AC1 | 0.770 | 0.898 | 0.746 |
| | AC2 | 0.933 | | |
| | AC3 | 0.881 | | |
| Personal Norm | PN1 | 0.608 | 0.866 | 0.623 |
| | PN2 | 0.800 | | |
| | PN3 | 0.889 | | |
| | PN4 | 0.532 | | |

*4.2. Goodness-of-Fit and Model Comparison*

This study conducts an analysis for goodness-of-fit with the results in Table 3. Table 3 shows that the research model's implementation of the data is fairly good. The chi-square/df (523.212/279) is 1.875, representing an acceptable fit. The comparative fit index (CFI), incremental fit index (IFI), and Tucker–Lewis index (TLI) are 0.949, 0.949, and 0.940 respectively, and all are bigger than 0.9. In addition, the root mean square error of approximation (RMSEA) is 0.062, or lower than 0.08, representing a great model fit [40].

**Table 3.** Index of reliability and validity.

| Model | $\chi^2$ | df | $\Delta \chi^2$ | CFI | IFI | TLI |
|---|---|---|---|---|---|---|
| Model 1: Hypothesized six-factor model (original model) | 523.212 | 279 | | 0.949 | 0.949 | 0.940 |
| Model 2: Alternative five-factor model (combination of PU and PEU) | 876.798 | 284 | 353.586 | 0.876 | 0.877 | 0.858 |
| Model 3: Alternative five-factor model (combination of AC and PN) | 625.053 | 284 | 101.841 | 0.928 | 0.929 | 0.918 |
| Model 4: Alternative four-factor model (combination of TR, PU and PEU) | 924.254 | 288 | 401.042 | 0.867 | 0.868 | 0.849 |
| Model 5: Alternative three-factor model (combination of TR, PU and PEU and combination of AC and PN) | 1440.020 | 293 | 916.808 | 0.760 | 0.761 | 0.733 |

Note: TR = Technology Readiness, PU = Perceived Usefulness, PEU = Perceived Ease of Use, INT = Intention to Adopt ES, AC = Awareness of Consequences, PN = Personal Norm.

This study also conducts an analysis of the chi-square test to verify that our research model is better than other models. (1) six-factor model (original model); (2) five-factor model (combination of perceived usefulness and perceived ease of use); (3) five-factor model (combination of awareness of consequences and personal norm); (4) four-factor model (combination of technology readiness, perceived usefulness, and perceived ease of use); (5) three-factor model (combination of technology readiness, perceived usefulness, and perceived ease of use and combination of awareness of consequences and personal norm). Based on Table 3, the results support that the six-factor model (original model) is better than the other alternative models.

### 4.3. Tests of Hypotheses

The results of the hypotheses appear in Figure 2. Based on TRAM, this study supposes that technology readiness positively influences perceived usefulness and perceived ease of use and proposes Hypotheses 1a and 1b. In Figure 2, we see that technology readiness not only has a positive relationship with perceived usefulness ($\beta = 0.822$, $t = 19.571$), but also influences perceived ease of use positively ($\beta = 0.897$, $t = 17.94$), thus supporting Hypotheses 1a and 1b. In this study, Hypotheses 2 and 3 describe that perceived usefulness and perceived ease of use have positive effects on consumers' intention of adopting ES. The results display that perceived usefulness has a positive effect toward adopting ES ($\beta = 0.276$, $t = 3.68$), while perceived ease of use also influences the intention to adopt ES positively ($\beta = 0.633$, $t = 8.554$).

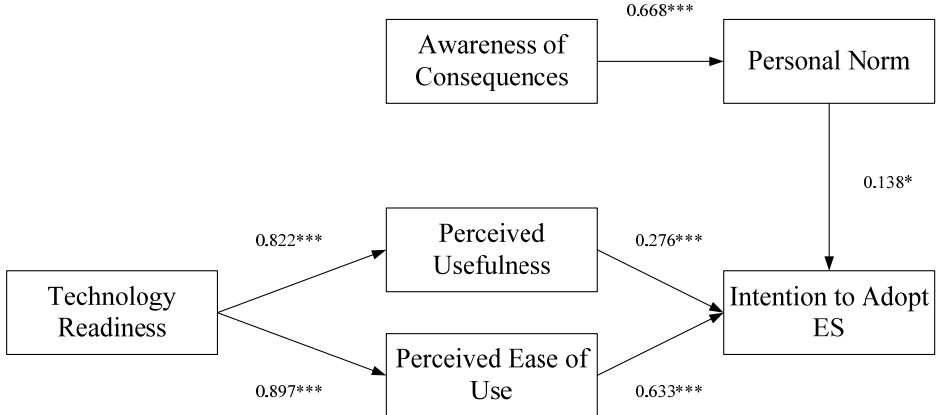

**Figure 2.** Results of the research framework. Notes: * $p < 0.05$, ** $p < 0.01$, *** $p < 0.001$.

Based on the norm activation model (NAM), this research proposes Hypotheses 4 and 5 and regards that awareness of consequences affects personal norm, and then personal norm positively influences consumers' intention toward adopting ES. We see that awareness of consequences affects personal norm significantly ($\beta = 0.668$, $t = 7.505$), and personal norm further positively influences intention toward adopting ES ($\beta = 0.138$, $t = 1.890$), thus supporting Hypotheses 4 and 5.

## 5. Discussion

With growing concerns about environmental problems and renewable energy, this study aims to explore crucial antecedents for improving consumers' intention for adopting ES instead of a conventional scooter. This study proposes an integrated model TRAM-NAM to examine the effects of technology readiness, perceived usefulness, perceived ease of use, awareness of consequences, and personal norm on the intention to adopt ES. The results show that technology readiness not only can, via perceive usefulness, influence the intention of adopting ES, but it also can, through perceived ease of use, affect the same intention. Moreover, personal norm mediates the positive relationship between awareness of consequences and intention for adopting ES. The results in this research provide an innovative reference for governments that want to deal with pollution and energy problems

through transportation. In addition, this study makes a contribution for academia and proposes TRAM-NAM, which is an integrated model, thus filling a gap in the literature of electric two-wheels.

*5.1. Theoretical and Practical Implications*

This study shows that people can, via two different processes, affect their intention of adopting EV at the same time. First, based on TRAM proposed by Lin et al. [23], the psychological traits will via cognitive dimensions predict people's acceptance of technologies [20]. Second, the norm activation model (NAM) developed by Schwartz [41] is interpreted as a process for the transfer of a moral norm into environmentally friendly actions. From the above, this study proposes an integrated model, denoted as TRA-NAM, and provides empirical tests to verify it. On one hand, a high TR score will influence people's perceived usefulness and ease of use positively and further enhance their intention at adopting ES. On the other hand, when people perceive the bad consequences from their transportation vehicle, it will arouse their moral obligation to do some friendly actions, thus increasing their intention to adopt ES. Along with the growing environmental concerns and advancements in technology in modern society, figuring out how to satisfy the technology and environmental demands of consumers at the same time is a great challenge. This study provides an integrated model to verify that, via two different processes, consumers can simultaneously make a decision.

This research discovers that perceived usefulness and perceive ease of use mediate the positive relationship between technology readiness and intention for adopting ES, implying their key roles. Governments that want to popularize ES can collaborate with the scooter industry to create excellent user experiences and reduce any disadvantages—for instance, provide appropriate technological support and service and set up enough charging stations for people to overcome their uncertainty or apprehension over using a new technology product. Moreover, the scooter industry might incorporate the factor of ease of use into the ES design process, which can benefit its competitive advantages.

The results also display that awareness of consequences can, through personal norm, positively influence the intention to adopt ES. Riding a conventional scooter can cause problems of carbon emissions and damage the environment, further exacerbating global warming and air pollution. Moreover, using a conventional scooter results in the problem of energy resource exhaustion. When people perceive that riding a conventional scooter can cause the above problems, it may spur their moral obligation and motivate them to exhibit a more environmentally friendly intention or behavior in order to decrease such negative consequences. Thus, when consumers make decisions about purchasing a scooter, instead of a conventional scooter, they may be more willing to choose ES. If governments want to encourage people to drive ES, they can use social media or traditional media to convey information about the negative outcomes of conventional scooters to motivate people to make decisions that are more environmentally friendly.

*5.2. Future Direction and Limitation*

Through the data collected in Taiwan, the results of this research show that technology readiness and awareness of consequences can, via personal norm, perceived usefulness, and ease of use, affect people's intention to adopt ES. Future research should collect data from different countries, such as China and Vietnam (whose citizens view scooters as their main form of transportation), to confirm the results. Moreover, many research studies adopt an experimental design to test the effect of different situations on green product purchase behaviors [42]. Therefore, future research can try to develop and create a condition that provides information about pollution that comes from conventional scooters, compare it to the condition without such information, and examine the effect of awareness of consequences.

Second, infrastructure plays an important role in developing EVs and greatly affects peoples' intention regarding the use of EVs [43]. Compared to traditional fuel vehicles,

charging stations for EVs are always the primary concern and might influence their intention to adopt EVs [22]. For this reason, undertaking the issue of charging infrastructure availability is necessary, and future research might try to explore the effect of infrastructure on consumers' behavior to adopt ES.

Third, future research might consider including other psychological variables in their analysis. For example, warm glow is defined as 'feeling good about one's self after engaging in pro-social behavior' [44]. Tezer et al. [42] emphasized the crucial psychological role of warm glow and showed that it mediates the relationship between green product and purchase behavior.

**Author Contributions:** Conceptualization, C.-W.H.; methodology, C.-W.H.; validation, C.-C.W.; formal analysis, C.-W.H.; investigation, C.-W.H. and C.-C.W.; writing—original draft preparation, C.-W.H. and C.-C.W.; funding acquisition, C.-W.H. All authors have read and agreed to the published version of the manuscript.

**Funding:** This research was funded by MOST, grant number 110-2410-H-011-002-.

**Institutional Review Board Statement:** Not applicable.

**Informed Consent Statement:** Informed consent was obtained from all subjects involved in the study.

**Data Availability Statement:** Not applicable.

**Conflicts of Interest:** The authors declare no conflict of interest.

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
