# Peer review of "Exploring Intention toward Using an Electric Scooter: Integrating the Technology Readiness and Acceptance into Norm Activation Model (TRA-NAM)"

_energies, doi:10.3390/en14216895_

Round 1

Reviewer 1 Report

I am not a specialist in statistical analysis. The presented mathematical apparatus and literature do not raise my concerns about the scientific reliability of the authors. I propose, after editorial correction, to accept this manuscript.

Extension:

Ad 1. It seems that the proposed integrated TRA-NAM model is accurate and promising. The concept/structure (Fig. 1) of the TRA-NAM model seems interesting and formally correct. 

The TRA-NAM model will better explain consumer behavior in relation to green vehicles / the limitation is their type - only scooters - although the authors explained why /. However, a question arises about the scope of application of the model and the application of the results. It should be noted that the authors proposed important, but still quite narrow/fragmentary / research - both in terms of the type of vehicle, as well as in terms of locality (place), as well as statistical representativeness. 

The series of working hypotheses H1-H5 should be well assessed; they are characterized by correctness, consistency, and completeness. The authors justified the selection of such hypotheses with the conclusions from the review of the results of the research conducted so far. The research base was well constructed. Of course, the nature of the research problem to be solved, as well as the H1-H5 hypotheses put forward, imply the choice of the research method - in particular, they imply a formal data collection process and a research procedure. 

Ad 2. I have no major objections to chapters 2 and 3 (Method), or to the results obtained (4. Results). I also have no objections to the mathematical statistics tools used - that is, first of all, the verification of the H1-H5 hypotheses, as well as the correlation analysis necessary here. 

However, I have two remarks which, although they do not depreciate the work as a whole, lower its value - mainly due to a "firstborn" error, namely a slightly opaque sampling frame (3.1), and the overlooked issue of "reliability and validation of results" (4.1).

Ad 3. What means reliability here (4.1)? a definition must be given. 

What correlation did the authors use here? Is it a correlation coefficient - what with what? It needs to be clarified. 

The presented results do not ensure the possibility of independent verification of the research by the reader. Only the results obtained by the authors are given, there is no data that would allow the procedure to be verified in the study. I propose to attach to the article the key data for the research. 

Note 1: (for 3.1). 

Not only the size/quantity of the random sample was not justified, but also its decomposition into sampling layers. The effective sample size was n = 227 items and we do not know how it relates to the size of the general population under study, or what statistical error was assumed. Thus, there is an understatement regarding the statistical representativeness of the research material. And this, after all, results in statistical credibility and possible applications of the obtained results (and conclusions in this regard). Therefore, the sampling frame has not been described in more detail, the main role of which is, inter alia, to ensure that each unit of the general population is taken into account and to avoid double testing of some units (in the case of complete surveys), and to allow each population unit to be included in the statistical sample. Admittedly, the authors quite vaguely ensure the reliability of pre-test procedures (Ad 3.1. Prior to the formal data collection process, this study conducted pre-test procedures to assure that our questionnaire is easy to understand and measures the research variables properly), it is a bit too little. Probably a more serious accusation is the lack of choice of the category of respondents in the questionnaire surveys "layers" of the statistical sample. 

Note 2: (for 4.1) 

Perhaps it was necessary to briefly explain how the correlations between the variables were measured (Tab. 1); this is important because the adopted other correlation indicators could give different (maybe small) values of the correlation between the variables of the examined model. Then such a model would simply be less (or not at all) useful. 

Author Response

Ad 1. It seems that the proposed integrated TRA-NAM model is accurate and promising. The concept/structure (Fig. 1) of the TRA-NAM model seems interesting and formally correct. 

The TRA-NAM model will better explain consumer behavior in relation to green vehicles / the limitation is their type - only scooters - although the authors explained why /. However, a question arises about the scope of application of the model and the application of the results. It should be noted that the authors proposed important, but still quite narrow/fragmentary / research - both in terms of the type of vehicle, as well as in terms of locality (place), as well as statistical representativeness. 

The series of working hypotheses H1-H5 should be well assessed; they are characterized by correctness, consistency, and completeness. The authors justified the selection of such hypotheses with the conclusions from the review of the results of the research conducted so far. The research base was well constructed. Of course, the nature of the research problem to be solved, as well as the H1-H5 hypotheses put forward, imply the choice of the research method - in particular, they imply a formal data collection process and a research procedure. 

Reply: Thanks for reviewer’s valuable advice. In order to solve the problem of locality, future research could collect data from different countries (whose citizens view scooters as their main form of transportation), we add this information in the part of future direction and limitation (p.8, line.350-352). Moreover, future research could try to include other type of vehicle to test the integrated model of TRA-NAM. Finally, this study adds information of data collection procedure (p.4).

Ad 2. I have no major objections to chapters 2 and 3 (Method), or to the results obtained (4. Results). I also have no objections to the mathematical statistics tools used - that is, first of all, the verification of the H1-H5 hypotheses, as well as the correlation analysis necessary here. 

However, I have two remarks which, although they do not depreciate the work as a whole, lower its value - mainly due to a "firstborn" error, namely a slightly opaque sampling frame (3.1), and the overlooked issue of "reliability and validation of results" (4.1).

Reply: Thanks for your kindly advice, this study adopts Pearson correlation analysis to confirm the significant correlation of each variable at first, and then uses SEM (structural equation modeling) to test the hypotheses (H1-H5) simultaneously.

This study also adds information to deal with issue about sampling frame (p4, line.173-177). In addition, we also supply information about reliability and validation (p5, line.247-249).

Ad 3. What means reliability here (4.1)? a definition must be given. 

What correlation did the authors use here? Is it a correlation coefficient - what with what? It needs to be clarified. 

Reply: Thanks for reviewer’s valuable advice, this study already adds definition of reliability in page 5 (line. 239-241). Moreover, we adopt Pearson correlation analysis to get correlation coefficient of each variable.

Note 1: (for 3.1). 

Not only the size/quantity of the random sample was not justified, but also its decomposition into sampling layers. The effective sample size was n = 227 items and we do not know how it relates to the size of the general population under study, or what statistical error was assumed. Thus, there is an understatement regarding the statistical representativeness of the research material. And this, after all, results in statistical credibility and possible applications of the obtained results (and conclusions in this regard). Therefore, the sampling frame has not been described in more detail, the main role of which is, inter alia, to ensure that each unit of the general population is taken into account and to avoid double testing of some units (in the case of complete surveys), and to allow each population unit to be included in the statistical sample. Admittedly, the authors quite vaguely ensure the reliability of pre-test procedures (Ad 3.1. Prior to the formal data collection process, this study conducted pre-test procedures to assure that our questionnaire is easy to understand and measures the research variables properly), it is a bit too little. Probably a more serious accusation is the lack of choice of the category of respondents in the questionnaire surveys "layers" of the statistical sample. 

Reply: Very thank for your valuable suggestion. This research tries to adopt some limitations to make sure the quality of the sample, such as excludes those respondents without a scooter license and those who have not ridden a scooter in the past two years. In addition, most of scooter’s drivers in Taiwan focus on the young population, and this information consists with the demographic information in this manuscript (the majority of respondents are between the ages of 18 and 25 (46.7%), followed by the ages between 26 and 35 (17.6%)). This study also adds information about data collection process (p.4, line.173-177).

Note 2: (for 4.1) 

Perhaps it was necessary to briefly explain how the correlations between the variables were measured (Tab. 1); this is important because the adopted other correlation indicators could give different (maybe small) values of the correlation between the variables of the examined model. Then such a model would simply be less (or not at all) useful. 

Reply: Thanks for reviewer’s valuable suggestion. This study adds information of correlations and revises the table 1 (p.5, line.236-238).

Reviewer 2 Report

The subject discussed by the authors is very important from the point of view of the problems arising from the operation of vehicles with conventional drive. This research is based on an accurate literature review as well as empirical research - a questionnaire study conducted by the authors. The aim of the study was clearly and precisely formulated and the theses put forward in the study were properly commented on. The advantage of the work is an appropriate description of the survey and then data verification. The results obtained were presented in a logical, proper manner and sufficiently commented on. The only suggestion from the reviewer is to stop using shortcuts. There are so many of them that the person reading the article may get confused. It can be annoying to constantly search through the text to decrypt the hash. 

Author Response

Review 2

The subject discussed by the authors is very important from the point of view of the problems arising from the operation of vehicles with conventional drive. This research is based on an accurate literature review as well as empirical research - a questionnaire study conducted by the authors. The aim of the study was clearly and precisely formulated and the theses put forward in the study were properly commented on. The advantage of the work is an appropriate description of the survey and then data verification. The results obtained were presented in a logical, proper manner and sufficiently commented on. The only suggestion from the reviewer is to stop using shortcuts. There are so many of them that the person reading the article may get confused. It can be annoying to constantly search through the text to decrypt the hash. 

Reply: Thanks for the reviewer’s valuable suggestion, we already revise shortcuts problems in this manuscript (not using too many abbreviations such as, TR, AC, and PN).

Reviewer 3 Report

The research focuses on the potential impact of technology on human consciousness, although, in my opinion, I do not consider that the classification of this publication is correct in "Energies".  Perhaps it would be better to link it to "sociology or market research". It's only a suggestion.

Author Response

Review 3

Comments and Suggestions for Authors

The research focuses on the potential impact of technology on human consciousness, although, in my opinion, I do not consider that the classification of this publication is correct in "Energies".  Perhaps it would be better to link it to "sociology or market research". It's only a suggestion.

Reply: Thanks for the reviewer’s kindly suggestion.
